# Acute Chyloperitoneum with Small Bowel Volvulus: Case Series and Systematic Review of the Literature

**DOI:** 10.3390/jcm13102816

**Published:** 2024-05-10

**Authors:** Teresa Sinicropi, Carmelo Mazzeo, Carmelo Sofia, Santino Antonio Biondo, Eugenio Cucinotta, Francesco Fleres

**Affiliations:** 1Section of General Surgery, Department of Human Pathology of the Adult and Evolutive Age “Gaetano Barresi”, University of Messina, Via Consolare Valeria, 98125 Messina, Italy; teresa.sinicropi@outlook.it (T.S.); dott.carmelomazzeo@gmail.com (C.M.); dott.santinoantoniobiondo@gmail.com (S.A.B.); ecucinot@unime.it (E.C.); 2Section of Radiological Sciences, Department of Biomedical Sciences and Morphological and Functional Imaging, University of Messina, Policlinico “G. Martino” Via Consolare Valeria 1, 98125 Messina, Italy; carm.sofia@tiscali.it

**Keywords:** acute chyloperitoneum, chylous ascites, small bowel volvulus, jejunoileal diverticula, acute chylous peritonitis, laparotomy

## Abstract

**Introduction:** Chyloperitoneum arises from lymph leakage into the abdominal cavity, leading to an accumulation of milky fluid rich in triglycerides. Diagnosis can be challenging, and mortality rates vary depending on the underlying cause, with intestinal volvulus being just one potential acute cause. Despite its rarity, our case series highlights chyloperitoneum associated with non-ischemic small bowel volvulus. The aims of our study include assessing the incidence of this association and evaluating diagnostic and therapeutic approaches. **Material and Methods:** We present two cases of acute abdominal peritonitis with suspected small bowel volvulus identified via contrast-enhanced computed tomography (CT). Emergency laparotomy revealed milky-free fluid and bowel volvulus. Additionally, we conducted a systematic review up to 31 October 2023, identifying 15 previously reported cases of small bowel volvulus and chyloperitoneum in adults (via the PRISMA scheme). **Conclusions:** Clarifying the etiopathogenetic mechanism of chyloperitoneum requires specific diagnostic tools. Magnetic resonance imaging (MRI) may be useful in non-emergency situations, while contrast-enhanced CT is employed in emergencies. Although small bowel volvulus infrequently causes chyloperitoneum, prompt treatment is necessary. The volvulus determines lymphatic flow obstruction at the base of the mesentery, with exudation and chyle accumulation in the abdominal cavity. Derotation of the volvulus alone may resolve chyloperitoneum without intestinal ischemia.

## 1. Introduction

Chyloperitoneum, also known as chylous ascites, arises from the leakage of chyle, a milky fluid rich in triglycerides (TG), into the peritoneal cavity [1,2]. The diagnosis of chylous ascites remains relatively uncommon, and its precise incidence remains elusive. Press et al. reported an incidence of 1 per 20,000 hospital admissions in the USA back in 1984 [3]. Associated with a mortality rate ranging from 40% to 70% and a correspondingly higher morbidity rate dependent on the underlying cause [4,5], chylous ascites presents a significant clinical challenge. Various etiologies have been implicated in its pathogenesis, categorized as either traumatic or non-traumatic [2,4,6]. However, the occurrence of chylous ascites alongside intestinal volvulus has been infrequently documented.

Volvulus ensues from the abnormal torsion of a segment of the bowel around its own mesentery, resulting in mechanical obstruction and presenting clinically as an acute abdomen. While volvulus of the colon, particularly involving the sigmoid colon (70–80%) and cecum (10–20%), is common in adults, volvulus of the small intestine is comparatively rare [7,8]. Recent studies indicate an annual incidence of small bowel volvulus in Western countries ranging from 1.7 to 5.7 per 100,000 adults [9]. In this context, we present a case series comprising two instances of chyloperitoneum associated with incomplete and non-ischemic small bowel volvulus.

## 2. Materials and Methods

### 2.1. Case Series

#### 2.1.1. Case 1

A 41-year-old male presented to our facility complaining of left flank pain accompanied by nausea and constipation persisting for 1 day. Notably, he had undergone a gastric mini-bypass procedure 5 years earlier. Upon clinical examination, he appeared afebrile and hemodynamically stable, with a slightly distended abdomen exhibiting generalized tenderness and normal bowel sounds. There were no signs suggestive of peritonism. Laboratory analyses yielded normal results, including lactate levels and ischemia indices (LDH or CPK).

Contrast-enhanced abdominal CT imaging revealed torsion of the mesenteric vascular axes in the left flank region, accompanied by imbibition and thickening of mesenteric fat and venous congestion. Additionally, free fluid was observed in the pelvis and between bowel loops. Subsequent exploratory laparotomy unveiled jejunoileal volvulus without intestinal ischemia, alongside milky free fluid accumulation between loops and within the pouch of Douglas (Figure 1). Notably, a whitish appearance at the mesenteric root indicated lymphatic infiltration. Surgical intervention comprised derotation of the affected intestinal loop, which exhibited vitality and lacked signs of vascular compromise, as well as peritoneal lavage with physiological solution. Furthermore, mesenteric lymph node excision for histological examination was performed. Intraperitoneal drains were strategically placed in the pouch of Douglas and left paracolic gutter, both exhibiting negative drainage by the second and fourth postoperative days (POD), respectively.

Analysis of intraperitoneal fluid revealed elevated triglyceride levels (1724 mg/dL), confirming its chylous nature, while microbiological examination yielded negative results. Histological examination of the excised lymph node excluded neoplastic etiology. The patient experienced an uneventful recovery, with resolution of chyloperitoneum, and was discharged 6 days post-surgery.

#### 2.1.2. Case 2

An 83-year-old male presented to our Emergency Department with complaints of abdominal pain in the lower quadrants and constipation persisting for 1 day. Pertinently, the patient disclosed engaging in painting activities while standing and bending his chest towards the floor during his leisure time. Medical history included colonic diverticulosis, Parkinson’s disease, and excision of a left lower limb melanoma 19 years prior. Physical examination revealed a distended abdomen with generalized tenderness.

Laboratory investigations indicated a reduced platelet count (126,000/mmc), along with elevated lactate dehydrogenase (LDH) (248 U/L) and creatine phosphokinase (CPK) (387 U/L) levels. Contrast-enhanced CT imaging exhibited twisting of the mesenteric vessels and small bowel around the mesentery, characterized by a “whirl sign” of the mesenteric vessels suggestive of volvulus, alongside free abdominal fluid accumulation (Figure 2).

On CT, a diffuse mesenteric fat stranding encountered in the context of a small bowel/mesenteric volvulus should raise suspicion of chyle infiltration of the mesentery.

Emergency exploratory laparotomy revealed a significant volume of milky fluid in all peritoneal recesses, non-ischemic jejunoileal volvulus, mesenteric root chylous infiltration, and numerous uncomplicated jejunal diverticula (diameter of about 4 cm) from about 20 cm after Treitz ligament for a distance of about 120 cm and numerous uncomplicated descending and sigmoid colon diverticula (Figure 3A,B). Surgical intervention comprised derotation of the affected intestinal loop, which displayed preserved peristalsis and lacked signs of vascular compromise, in addition to peritoneal lavage. Intraperitoneal drains were placed in the pouch of Douglas and perigastric site, both removed on the fourth and fifth POD, respectively.

Analysis of intraperitoneal fluid confirmed elevated triglyceride levels (2488 mg/dL), consistent with its chylous nature, while microbiological examination yielded negative results. The postoperative course was uneventful, with resolution of chyloperitoneum, and the patient was discharged on the seventh POD.

## 3. Systematic Review

We conducted a systematic review of the literature to comprehensively assess several key aspects: the true incidence of chyloperitoneum associated with small bowel volvulus, elucidation of the underlying pathophysiological mechanism for this association, and identification of the most effective diagnostic and therapeutic approaches reported in the existing literature.

To achieve this, we conducted a thorough search of the PubMed database spanning from 1958 to 31 October 2023, utilizing the keywords “*chyloperitoneum*”, “*chylous ascites*”, “*small bowel volvulus*”, and “*adult*”. We have extended our research to EMBASE and MEDLINE, but unfortunately, we have not found additional papers.

Our search yielded a limited pool of relevant cases, with only 15 previously reported instances of small bowel volvulus concomitant with chyloperitoneum in adults, as depicted in the PRISMA scheme (Figure 4). Notably, two non-English articles were excluded from our analysis due to the risk of bias caused by incorrect translation, leaving us with 13 cases for detailed examination (Table 1).

## 4. Results of Systematic Review

It is imperative to note that in two cases, intestinal volvulus was detected on CT scans but not confirmed during surgical exploration. Moreover, both patients presented symptoms subsequent to vigorous physical exertion, suggesting a potential spontaneous resolution of torsion persisting long enough to produce chyloperitoneum.

Table 1 encompasses our two patients alongside the thirteen identified cases. The average age among the 15 patients was 59 years, with 3 women and 12 men represented in the cohort. Despite significant intra-abdominal chylous fluid accumulation, metabolic abnormalities necessitating correction were not observed, likely attributable to the acute onset of symptoms and prompt surgical intervention. Abdominal pain was universally reported among all cases, likely stemming from both the volvulus and the presence of chyloperitoneum.

Diagnostic modalities predominantly relied on CT imaging, with contrast-enhanced scans utilized in eight cases. Exploratory laparotomy was the primary surgical approach in 11 cases, while laparoscopy was performed in 4 cases, with 2 instances necessitating conversion to open procedures.

Incidentally discovered during surgery, chyloperitoneum was managed by derotation of the affected loop without necessitating intestinal resection. Notably, the presence of small bowel volvulus was not confirmed during surgery in only two cases, whereas large small bowel diverticula (4 cm) were incidentally identified in one patient (case 2).

Intra-abdominal drains were placed in six cases, deemed unnecessary in three cases, and not reported in six cases. Patients were discharged, on average, on the sixth POD, with hospitalization durations ranging from one to sixteen days post-surgery, and were unreported in two cases.

Triglyceride content in intraperitoneal fluid was evaluated in all cases, with values explicitly mentioned in nine instances, averaging 1000 mg/dl, with a range of 332 to 2488 mg/dl. Cytological testing was conducted in six cases, while microbiological examination of intra-abdominal fluid was performed in nine cases. Histological examination of lymph nodes extracted during surgery was requested in three cases, yielding negative results for malignancy in all instances.

Notably, no postoperative complications, hospital deaths, or chyloperitoneum recurrences following volvulus resolution were reported in the cases examined.

## 5. Discussion

Chylous ascites, characterized by the accumulation of triglyceride-rich fluid in the abdominal cavity, can arise from several mechanisms, as proposed in the literature [5].

Exudation of chyle secondary to lymphatic obstruction at the root of the mesentery or cisterna chyli, often associated with conditions such as malignant infiltration or intestinal volvulus, frequently resulted in protein-losing enteropathy.

There was direct leakage of chyle through a lymphoperitoneal fistula, typically following trauma or surgical injury causing disruption of the retroperitoneal lymphatic vessels.

Exudation of chyle through retroperitoneal lymphatics, with or without visible fistula, was commonly observed in congenital lymphangiectasia or thoracic duct obstruction.

Various analyses can be performed on intraperitoneal fluid, including assessment of gross appearance, cell count, chemical composition, cytology, and microbiological examination [4]. Currently, a triglyceride content exceeding 200 mg/dL serves as the diagnostic threshold for chylous ascites [2].

Chyloperitoneum has multifarious etiologies [2,4,5,6], with abdominal malignancy and cirrhosis being predominant in Western countries, while infectious causes like tuberculosis and filariasis are prevalent in Eastern and developing regions [2]. An uncommon yet noteworthy cause is small bowel volvulus [11,12,13,14,15,16,17,18,19,20,21].

Volvulus, characterized by the rotation of the mesentery in either a clockwise or counterclockwise direction, is rare in the small intestine. It can be primary, occurring in a healthy abdomen due to factors such as dietary habits. This may include the consumption of a large quantity of poorly digestible food after a period of fasting, which could induce intense intestinal peristalsis, potentially leading to volvulus. Alternatively, the condition can have a secondary origin, precipitated by predisposing conditions such as anatomical anomalies (malformations, malrotations, and non-rotations) or postoperative adhesions [7,8,22]. Venous and lymphatic obstructions ensue, leading to lymphatic congestion, chyle exudation, and subsequent chyloperitoneum [5,18].

It is noteworthy that small bowel diverticula, as observed in our patient (case 2), can predispose to volvulus. Jejunoileal diverticula, characterized by mucosal and submucosal outpouchings originating from the antimesenteric border of the intestine, have an incidence ranging from 0.5% to 2.3%, with multiple diverticula detected in 77% of cases. Middle-aged-to-elderly men are predominantly affected [21,22,23,24]. Diagnosis of diverticula is challenging due to non-specific clinical manifestations, necessitating imaging studies, particularly CT scans with oral contrast. Notably, larger diverticula (>3 cm) and the numbers pose an increased risk of small bowel volvulus [25]. As evidenced by the literature, one or more large jejunoileal diverticula are found in 35% of patients with small bowel volvulus. The existence of adhesions around the diverticulum and the possibility of triggering and maintaining an abnormal rotation of the intestine are at the base of volvulus development [26,27,28,29,30].

CT imaging plays a pivotal role in diagnosing volvulus, manifesting as a whirlpool sign indicating the twisting of mesenteric vessels and the small bowel around the mesentery [29,30]. Endovenous contrast aids in assessing bowel vascular integrity.

The whirlpool sign may not be as apparent if the axis of rotation is not perpendicular to the transverse scanning plane. The CT of the abdomen is a useful tool with which to identify abdominal masses and lymph nodes, but it is often difficult to distinguish the nature of the intra-abdominal free fluid [31].

However, distinguishing the nature of intra-abdominal free fluid can be challenging. In cases of chylous fluid, CT scans performed after the patient assumes a supine posture may reveal a fat layer due to differential displacement of lipid molecules, aiding in diagnosis [4].

Alternative modalities for lymphatic system assessment include lymphoscintigraphy, lymphangiography, and MRI, particularly useful in non-emergent scenarios for evaluating retroperitoneal abnormalities, fistulas, or lymphatic duct leakage and assessing thoracic duct patency. MRI lymphography is especially adept at delineating anatomical variations like giant cisterna chyli or abnormal lymphatic dilatation, distinguishing them from fluid collections observed on CT scans [32,33].

The pathogenesis of pain in chyloperitoneum remains elusive, possibly arising from the direct contact of chyle with peritoneal serosa or the stretching of mesenteric serosa and retroperitoneum [34].

Chyloperitoneum is believed to develop when applied pressure completely occludes the lymphatic vessel with a lower venous blood flow yet fails to obstruct arterial circulation. Consequently, if the superior mesenteric artery is entirely blocked, chylous ascites cannot be produced. On the other hand, chylous ascites can be produced when light pressure is applied near the root of the superior mesenteric vessels, causing a complete obstruction of lymphatic vessels. Given that lymph is produced from venous blood, lymph leakage can occur due to the maintenance of arterial blood flow. Therefore, the presence of chylous ascites could suggest a high probability of intestinal preservation, thanks to the conservation of blood flow [11]. Indeed, in all analyzed cases, including our two cases, no intestinal necrosis was observed, and bowel resection was not necessary.

While minimally invasive surgery has revolutionized patient recovery and complication rates, the priority remains patient safety and the expeditious resolution of acute issues, often necessitating a laparotomic approach, particularly in cases of distended and volvulated bowel (Table 1: laparotomy in 13/15 patients).

The careful handling of the distended bowel should be performed very cautiously to avoid iatrogenic complications (ex. perforation or vascular lesions), especially in cases of a distended and volvulated bowel.

Surgical intervention typically involves the derotation of the volvulus, peritoneal lavage, and meticulous bowel inspection. Intestinal resection with or without primary anastomosis may be warranted in cases of vascular compromise.

## 6. Conclusions

Small bowel volvulus presents as a rare but clinically significant cause of chyloperitoneum, necessitating prompt treatment. The presence of small bowel diverticula and strenuous physical effort may increase the risk of volvulus.

We believe that the volvulus caused the twisting of the lymphatic vessels in the root of the mesentery and their obstruction due to low pressures in this system. The mesentery became engorged with chyle, resulting in white staining. The obstruction determines the increase in pressure inside the vessels and the exudation of the chyle with the appearance of the chyloperitoneum [15].

Timely derotation of the volvulus and peritoneal lavage facilitate resolution of chyloperitoneum in the absence of vascular distress. Subsequent evaluation with MRI and other modalities may be warranted to assess lymphatic system abnormalities following volvulus resolution.

## Figures and Tables

**Figure 1 jcm-13-02816-f001:**
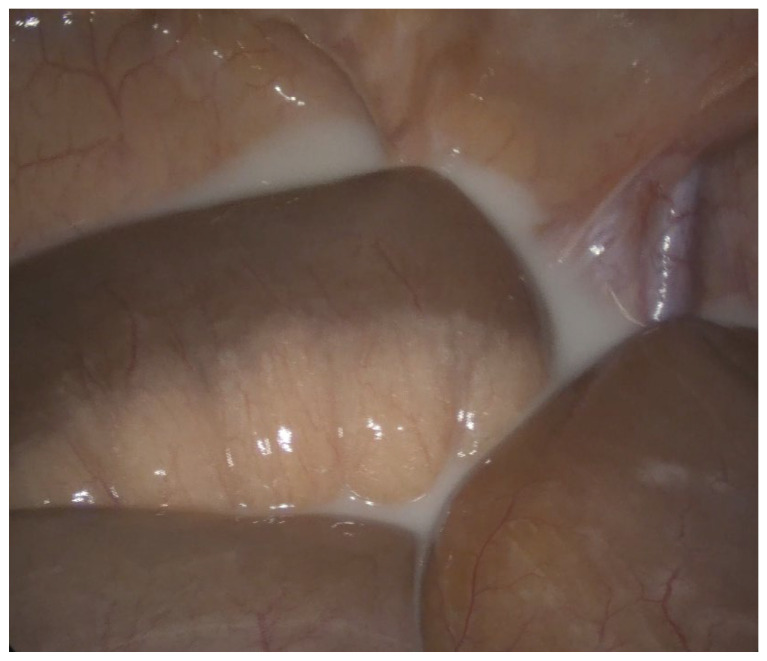
Chyloperitoneum—free fluid in the pouch of Douglas.

**Figure 2 jcm-13-02816-f002:**
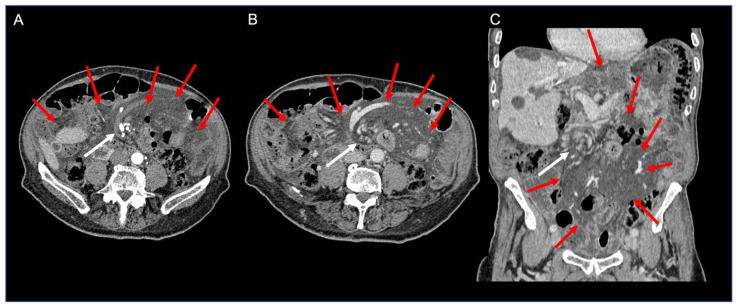
Contrast-enhanced CT axial arterial (**A**) and venous phase (**B**) images and coronal venous phase (**C**) image of the abdomen showing diffuse fat stranding of the mesentery (red arrows), containing multiple nodes. Note the whirl-like rotation of the mesentery and mesenteric veins around the superior mesenteric artery (“whirlpool sign”—white arrows).

**Figure 3 jcm-13-02816-f003:**
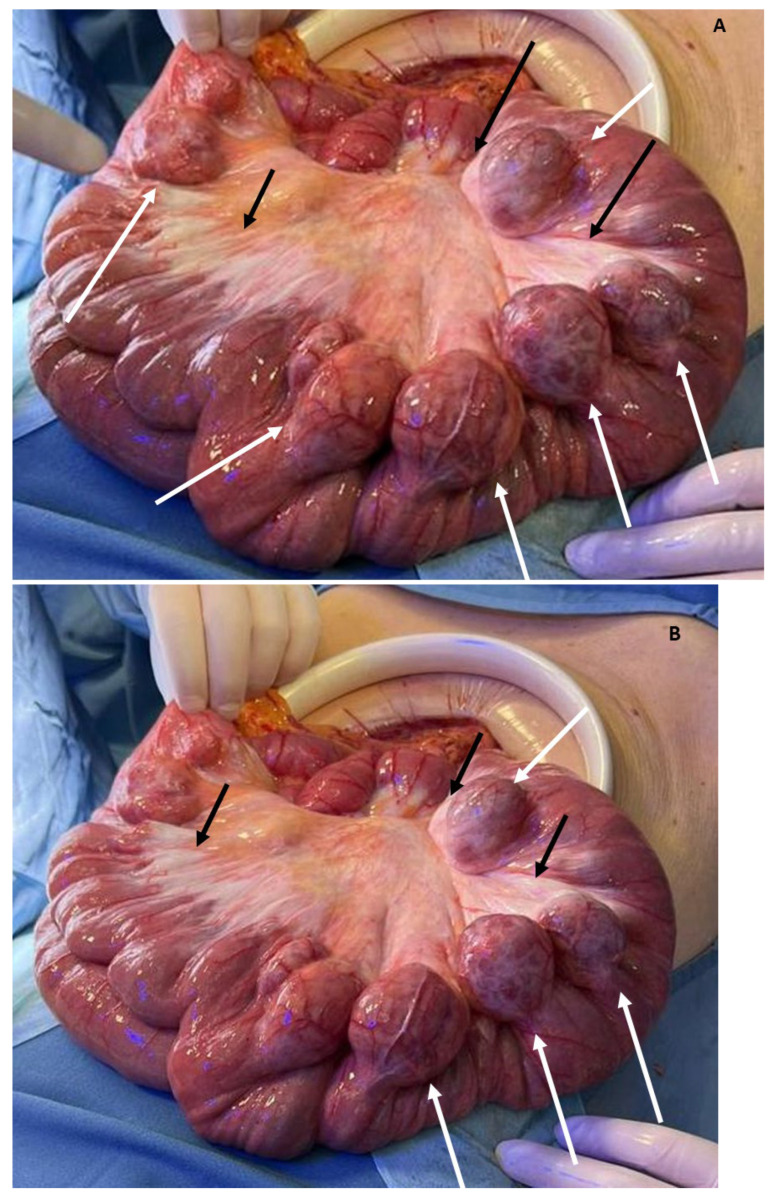
(**A**,**B**) Mesenteric chyle infiltration at the root of mesentery in small bowel volvulus (black arrows); large jejunal diverticula (white arrows).

**Figure 4 jcm-13-02816-f004:**
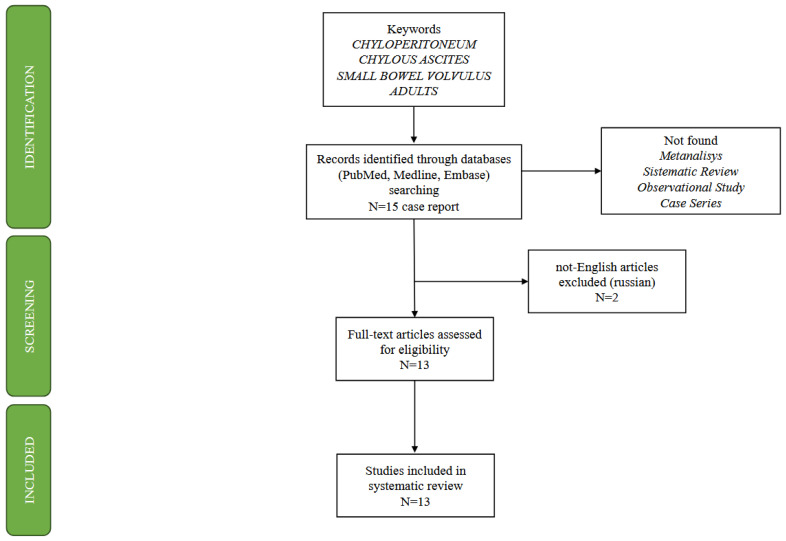
PRISMA Scheme using the keywords “*chyloperitoneum*”, “*chylous ascites*”, “*small bowel volvulus*”, and “*adult*”.

**Table 1 jcm-13-02816-t001:** Summary of acute chylous peritonitis with small bowel volvulus reported in the literature with references and our two cases. All identified articles were case reports.

Refs.—Year of Publication	Gender	Age	Main Symptoms	Previous Surgical Interventions	Abnormal Lab Results	Diagnostic Modality	Management	Presence of Volvulus during the Surgery	Small Bowel Diverticula	Intestinal Resection	Drainage (Number)	Day of Discharge	Triglycerides (mg/dL)	Cytological Examination	Microbiological Examination	Histological Examination of Lymph Nodes
Gritsiuta, A.I. [1]—2022	M	44	epigastric pain	umbilical hernia repair		contrast CT	exploratory laparoscopy converted to open	no	no	no	no	1st POD	1451			
Tewari, N. [10]—2012	M	38	abdominal pain with nausea and vomiting		WBC 20,100	contrast CT	exploratory laparotomy	no	no	no	yes	5th POD	NM			
Hayama, T. [11]—2017	M	70	distension and abdominal pain		CPK 294	abdominal X-ray, contrast CT	exploratory laparotomy	yes	no	no	NM	10th POD	332			
Koh, Y. [12]—2013	M	19	abdominal pain with vomiting		WBC 12,880	CT	exploratory laparotomy	yes	no	no	yes	NM	504	negative	negative	negative for malignacy
Leaning M [13]—2021	F	79	epigastric pain		WBC 13,400	CT	exploratory laparotomy	yes	no	no	NM	7th POD	770	negative	negative	
Akama, Y. [14]—2016	M	85	distension and abdominal pain	subtotal gastrectomy with roux-en-y reconstruction for early gastric cancer	WBC 8700, CRP 16.2	contrast CT	exploratory laparotomy	yes	no	no	yes (2)	16th POD	642	negative	negative	
Pai, A. [15]—2014	M	67	cramping abdominal pain and constipation	sigmoid colectomy for volvulus		contrast CT	exploratory laparoscopy converted to open	yes	no	no	no	3rd POD	NM		negative	
Vishnoi, V. [16]—2018	M	80	abdominal pain with nausea and vomiting	laparoscopic bilateral inguinal hernia repair	WBC 18,100	abdominal X-ray, CT	exploratory laparotomy	yes	no	no	no	5th POD	NM	negative	negative	negative for malignacy
Pengelly, S. [17]—2012	F	85	cramping abdominal pain and vomiting	laparoscopic cholecystectomy		CT	exploratory laparotomy	yes	no	no	NM	4th POD	NM			
Murugan, K. [18]—2008	M	44	cramping abdominal pain	partial esophago-gastric resection for boerhaave’s syndrome		abdominal X-ray, contrast CT	exploratory laparotomy	yes	no	no	NM	NM	NM			
Crang, N. [19]—2022	M	80	abdominal pain with nausea	right inguinal hernia repair	WBC 10,240, LAC 4.5	CT	exploratory laparotomy	yes	no	no	yes	9th POD	NM	negative	negative	
Hsu, C.H. [20]—2023	F	37	abdominal pain with nausea and vomiting	gastric clipping with proximal jejunal bypass		CT	exploratory laparoscopy	yes	no	no	NM	5th POD	726	negative	negative	
Gupta, S. [21]—2023	M	32	abdominal pain with nausea and vomiting		WBC 13,800	CT	exploratory laparoscopy	yes	no	no	NM	2nd POD	360			
Case 1—2020	M	41	left flank pain with nausea and costipation	mini gastric bypass		contrast CT	exploratory laparotomy	yes	no	no	yes (2)	6th POD	1724		negative	negative for malignacy
Case 2—2022	M	83	abdominal pain and costipation	lower limb melanoma excision	PLT 126,000, LDH 248, CPK 387	contrast CT	exploratory laparotomy	yes	yes	no	yes (2)	7th POD	2488		negative	

WBC: white blood cells (mmc); CPK: creatine phosphokinase (U/L); LAC: lactate (mmol/L); CPR: c-reactive protein (mg/L); PLT: platelets (mmc); LDH: lactate dehydrogenase (U/L); CT: computed tomography; POD: postoperative day; NM: not mentioned.

## Data Availability

Data is contained within the article.

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
