# Peer review of "Acute Chyloperitoneum with Small Bowel Volvulus: Case Series and Systematic Review of the Literature"

_jcm, 2024, doi:10.3390/jcm13102816_

Round 1

Reviewer 1 Report

Comments and Suggestions for Authors

I have reviewed in detail the paper entitled: “Acute chyloperitoneum with small bowel volvulus: case series and systematic review of the literature”. In this article, the authors presented two cases of the disease and conducted a bibliographic review of the cases reported over more than 60 years. I comment follow:

1.       I consider that the article addresses an interesting topic, with a very rare condition; however, the authors included two detailed cases of the disease and subsequently the review of the literature. I consider that the authors should only carry out the systematic review (including their cases) and publish their findings in another article as a case report.

2.       The authors mention Table 1 and table 2 in the text but did not include them.

3.       Figure 2 is incomplete

4.       Supplementary tables and figures are not mentioned in the main text

5.       The appointments do not correspond to the format requested by JCM. They should check the style. In addition they account for errors for example: As they cite: “Chyloperitoneum or chylous ascites is caused by the leakage of chyle, which is a milk 37 fluid rich in triglycerides (TG), into the peritoneal cavity” [1], [2].

A. I. Gritsiuta, A. Bracken, J. Lara-Gutierrez, and W. N. Gilleland, ‘Sit-ups and emergency abdominal surgery: a  rare case of intestinal volvulus and resultant chylous ascites incited by abdominal exercises’, Journal of Surgical Case Reports, vol. 2022, no. 4, p. rjac155, Apr. 2022, doi: 10.1093/jscr/rjac155.

As should be cited according to the JCM: “Chyloperitoneum or chylous ascites is caused by the leakage of chyle, which is a milk 37 fluid rich in triglycerides (TG), into the peritoneal cavity” [1,2].

Gritsiuta, A.I.; Bracken, A.; Lara-Gutierrez, J.; Gilleland, W.N. Sit-ups and emergency abdominal surgery: a rare case of intestinal volvulus and resultant chylous ascites incited by abdominal exercises. J Surg Case Rep 2022, 2022, rjac155, doi:10.1093/jscr/rjac155.

Author Response

First of all, we want to thank you for your suggestions.

  1. Thank you for having appreciated our article on a very rare condition. We have thought about publishing two papers (the systematic review and a case series separately), but as you know case reports and case series have lost importance. So, we believe that in this way it can have more diffusion between readers.
  2. That was a mistake, we have corrected there is only table 1.
  3. Figure 2 is complete, but it was out of the borderline. We have corrected it.
  4. In the main text the table and figures are mentioned.
  5. We have corrected the references’ style, according to the rules of the JCM.

Reviewer 2 Report

Comments and Suggestions for Authors

The authors reported 2 cases of chylous ascites due to small intestinal volvulus. It is a rare condition, however, its importance comes from the differential diagnosis with causes of intestinal obstruction and acute abdomen. There are a few comments:

What are the surgical steps done to prevent the recurrence of the volvulus?

Keywords: triglycerides, laparoscopy ------> To be removed; it does not add value.

Line 43: traumatic and not causes -------> Bad expression, to be changed.

Line 48-49: Volvulus of the small intestine, however, is relatively rare in contrast ----------------> In contrast, volvulus.......

Line 60: lactate and ischemia 60 indices -------> What do you mean by ischemic indices; write in detail.

Line 81:colic ----------> colonic.

Line 87: Is the whirl sign of the CT examination pathognomonic for the volvulus? If it is not present, does this mean the absence of volvulus?

Results:

Line 142, 143:  The triglyceride content of the intraperitoneal fluid was assessed in all patients; however, its value was mentioned in only 9 cases. The mean value was 1000 mg/dl, with a minimum value of 332 mg/dl and a maximum of 2488 mg/dl--------> What is the value of determining the triglyceride level in such situations???! Is the white color of the fluid is not enough for diagnosis of chylous fluid in contrast to pus which is thick fluid. Did you analyze the fluid intraoperatively or later? Do you use ether in the operative theatre to prove the fatty nature of the fluid?

Comments on the Quality of English Language

Minor corrections.

Author Response

What are the surgical steps done to prevent the recurrence of the volvulus?

Keywords: triglycerides, laparoscopy ------> To be removed; it does not add value.  They are removed.

Line 43: traumatic and not causes -------> Bad expression, to be changed.  We have corrected them.

Line 48-49: Volvulus of the small intestine, however, is relatively rare in contrast ----------------> In contrast, volvulus.......     We have corrected.

Line 60: lactate and ischemia 60 indices -------> What do you mean by ischemic indices; write in detail. Patients had normal level of LDH and CPK.

Line 81:colic ----------> colonic. We have correct it, thanks.

Line 87: Is the whirl sign of the CT examination pathognomonic for the volvulus? If it is not present, does this mean the absence of volvulus? Excuse me, I cannot well understand your question. In line 86-87 we wrote: “The contrast-enhanced CT showed a twisting of the mesenteric vessels and small bowel around the mesentery with mesenteric vessels creating the whirl sign suggestive of volvulus and free abdominal fluid (Fig 2).” This is the description of our findings in patient number 2. We have identified the whirl sign at CT and the volvulus was confirmed during surgery. As you know the whirl sing is a specific sign at CT of volvulus. The “whirlpool sign” is a term used in radiology to describe the appearance of twisting or rotation of an anatomical structure. Specifically, in the context of the mesentery, the whirlpool sign is observed when the intestine rotates around its mesentery, creating a spiral appearance of the mesenteric structures and superior mesenteric veins around the superior mesenteric artery. This sign is often associated with intestinal volvulus. The whirlpool sign is a visual clue that can aid in the diagnosis of specific conditions, but its absence does not necessarily rule out the presence of those conditions.

Results:

Line 142, 143:  The triglyceride content of the intraperitoneal fluid was assessed in all patients; however, its value was mentioned in only 9 cases. The mean value was 1000 mg/dl, with a minimum value of 332 mg/dl and a maximum of 2488 mg/dl--------> What is the value of determining the triglyceride level in such situations???! Is the white color of the fluid is not enough for diagnosis of chylous fluid in contrast to pus which is thick fluid. Did you analyze the fluid intraoperatively or later? Do you use ether in the operative theatre to prove the fatty nature of the fluid?

Thank you that is an interesting observation: we have found a milky free fluid during surgery, so we have performed in the same time all histologic, cytologic, microbiological and triglyceride exams. In the papers we have reviewed it were not clearly reported, although we understand that the authors have performed the analysis during or immediately after surgery due to the findings of the white color of the fluid.

Reviewer 3 Report

Comments and Suggestions for Authors

Dear authors, I had the pleasure to review your case series with systematic review of the literature on acute peritoneum with small bowel volvulus. Unfortunately the scientific and formal outline of the paper does not match with the interest of the topic analysed. In the case series a lot of major concerns on english language and on format (punctuation). The systematic review of the literature displays lots of concerns, even if it is stated that follows PRISMA statement. Non English studies should not be excluded. The PRISMA checlikst should be implemented. In a systematic review a table showing baseline study characteristics should be implemented. At least 3 databases should be searched and the PRISMA flowchart of screening and inclusion should therefore be presented. 

Comments on the Quality of English Language

Extensive editing of english and punctuation

Author Response

Thank you for your review and suggestions.

We have reviewed language mistakes by a native mother language.

Format and punctuation were both reviewed.

We believe you can understand that we were forced to exclude non-English papers due to the risk of bias caused by incorrect translation.

The PRISMA checlikst should be implemented. In a systematic review a table showing baseline study characteristics should be implemented.  All studies, that we have analysed, are case report, this is the first systematic review to our knowledge.

At least 3 databases should be searched and the PRISMA flowchart of screening and inclusion should therefore be presented. 

We have conducted our research on PubMed, and after your suggestion, we have extended it to EMBASE and MEDLINE. Unfortunately, we have not found additional papers

Round 2

Reviewer 2 Report

Comments and Suggestions for Authors

Can be published in the present shape.